# Assessing the influence of the health system on access to cervical cancer prevention, screening, and treatment services at public health centers in Addis Ababa, Ethiopia

Kemal Hussein[1]*, Gilbert Kokwaro[1], Francis Wafula[1], Getnet Mitike Kassie[2]

**1** Institute of Healthcare Management, Strathmore University, Nairobi, Kenya, **2** International Institute for Primary Healthcare – Ethiopia (IPHC-E), Addis Ababa, Ethiopia

* kemalahmedfenet@gmail.com

**Data Availability Statement:** All relevant data are within the manuscript and its Supporting information files.

## Abstract

### Background

Cervical cancer is the second leading cause of cancer death among Ethiopian women. This study aimed to assess the influence of the health system on access to cervical cancer prevention, screening, and treatment services at public health centers in Addis Ababa, Ethiopia.

### Methods

This study used a cross-sectional survey design and collected data from 51 randomly selected public health centers in Addis Ababa. Open Data Kit was used to administer a semi-structured questionnaire on Android tablets, and SPSS version 26 was used to analyze the descriptive data.

### Results

In the study conducted at 51 health centers, cervical cancer prevention and control services achieved 61% HPV vaccination for girls, 79% for cervical cancer awareness messages, 80% for precancer lesion treatment, and 71% for cervical screening of women. All health centers were performing cervical screening mostly through visual inspection with acetic acid due to the inconsistent availability of HPV DNA tests and the lack of Pap smear tests. In 94% of health centers, adequate human resources were available. However, only 78% of nurses, 75% of midwives, 35% of health officers, and 49% of health extension workers received cervical cancer training in the 24 months preceding the study. Women had provider choices in only 65% of health centers, and 86% of the centers lacked electronic health records. In 41% of the health centers, the waiting time was 30 minutes or longer. About 88% and 90% of the facilities lacked audio and video cervical cancer messages, respectively.

**Funding:** The author(s) received no specific funding for this work.

**Competing interests:** The authors have declared that no competing interests exist.

## Conclusion

This study revealed that the annual cervical cancer screening achievement was on track to fulfill the WHO's 90-70-90 targets by 2030. We recommend that decision-makers prioritize increasing HPV vaccination rates, enhancing messaging, reducing wait times, and implementing electronic health records to improve access to cervical cancer services in Addis Ababa.

## Introduction

Though cervical cancer is a preventable disease and curable (with early detection, timely diagnostic follow-up, and effective treatment) it continues to be a public health problem resulting in the premature death of women mainly in low-income countries globally [1]. In 2020, for instance, globally, 604, 127 new cervical cancer cases were reported with 341,831 deaths of which 117,316 incidences and 76, 745 deaths occurred in Africa [2]. Cervix uteri cancer accounted for 23.3% of all new cases of cancer in females in Sub-Saharan Africa (SSA) in 2020 [3]. In 2020, the age-standardized incidence rate (ASIR) for Eastern Africa was 40 cases per 100,000 women [2]. This is ten-fold higher than the incidence rate of lower than four per 100,000 women set by the WHO Global Strategy [1]. The pooled incidence proportion of estimates of high-risk human papillomavirus (HPV) infection among SSA women was 34% [4].

Each year, hospital data in Ethiopia revealed over 150,000 new cases of cancer, which account for 4% of all deaths [5]. In 2020 in Ethiopia, cervical cancer was the second most prevalent type of cancer among women, affecting 7,445 women and responsible for 5,338 deaths [6]. The ASIR was 24.6 per 100,000 Ethiopian females in 2019 [7]. The country had 36.9 million women aged 15 years and older who were at risk of developing cervical cancer, and an estimated 3.8% of women in the general population harbor cervical HPV16/18 infection [8]. In Addis Ababa, the most common cancers in women were breast (33%), and cervix uteri cancer (14.3%) [9, 10].

According to the 2017 WHO report policymakers and partners at the national and international levels paid insufficient attention to cancer [11]. Ethiopia regretfully also devoted little attention to cancer despite it being a significant public health issue [5, 9]. To prepare the way for the eradication of cervical cancer in the twenty-first century, WHO urges nations to achieve the 90-70-90 cervical cancer targets by 2030. At the age of 15, 90% of girls take the HPV vaccine in full, 70% of women undergo cervical screening at the age of 35 and 45, and 90% of women with precancerous lesions or invasive cancer receive treatment [1]. To combat cancer, the Ethiopian National Cancer Control Plan (NCCP) was put into effect in 2015 [9]. However, the healthcare system in the country still had to contend with issues like low levels of public awareness about cancer, a shortage of diagnostic and treatment facilities, a shortfall of oncology specialists, and a poor referral system [12]. Only 9% of health facilities offered diagnosis and treatment for cervical cancer in 2018 [13]. In 2014, just 8% of facilities had guidelines for the diagnosis and treatment of cancer, and 4% of those facilities had staff who had undergone cancer in-service training [14]. The estimated pooled prevalence of cervical cancer screening service utilization was 5.47% [15]. In 2020, there were still reports of high cervical cancer morbidity and mortality rates [6].

Our study aimed to assess the health system's influence on access to cervical cancer prevention, screening, and treatment services at public health centers in Addis Ababa. The study took into account components of the earlier conceptual frameworks to examine the health system's

"inputs" and "outputs" related to services for cervical cancer [16, 17]. "Quality healthcare in Ethiopia" calls for comprehensive care that is timely, affordable, efficient with its use of resources, effective, safe, and patient-centered [18]. Similarly, in this study, a "quality cervical cancer service" refers to how well the desired health outcomes (target achievement, timeliness, patient-centeredness, availability, and equity) are improved by providing women with services for prevention, screening, and treatment. While "equity" seeks to provide high-quality health-care services to all people without distinction based on their geographical location, gender, income, or disabilities [19].

## Materials and methods

### Study settings and period

Ethiopia's three-tiered health service delivery system includes primary, secondary, and tertiary-level care. The primary healthcare institutions (health centers), which each provide services to up to 40,000 people in urban settings, are at the bottom of the tier structure [20]. The location of this study was Addis Ababa, the capital city of Ethiopia, which has 126 woredas (districts) and 11 sub-cities. According to UN population forecasts, Addis Ababa's metro area had a population of 5,228,000 in 2022 [21]. To uncover problems in a health system setup with reasonably enough resources and to suggest workable solutions that may be extended to other places while taking contextual variables into account, this urban scenario was selected. The study's data collection period was from July 15, 2022, until August 31, 2022.

### Study design and participants

This study employed a cross-sectional survey design to assess the influence of the health system on access to prevention, screening, and treatment of cervical cancer at public health centers in Addis Ababa. The study population was 90 public health centers actively providing cervical cancer prevention, screening, and treatment services. Study participants comprised 51 cervical cancer focal persons appointed by the 51 health centers chosen for the study. Nurses, midwives, and health officers who supervise, manage, and coordinate cervical cancer services are the focal individuals in Addis Ababa's health center settings.

### Sample size and sampling procedure

The sample size for health centers was determined using $n_0 = z^2 * p * q / e^2$ [22]. Where: $n_0$ = sample size, z = 1.96 for a confidence level ($\alpha$) of 95%, p = proportion (0.09) [13], q = 1-p (0.91), e = margin of error (5%). Thus, the initial sample size ($n_0$) consisted of 126 facilities. Since the population size is small to the sample size, the sample size was adjusted using $n = (n_0/(1+(n_0-1)/N)))$ [22]. Where n is the adjusted sample size and N is the population size. A proportionate number of 51 health centers were randomly selected using Microsoft Excel out of a total of 90 public health centers actively providing cervical cancer services in 11 sub-cities. The list was obtained from the Addis Ababa City Administration Health Bureau. Additionally, after considering the "cervical cancer focal person position" in the health centers, 51 healthcare workers (nurses, midwives, and health officers) were chosen for face-to-face interviews using a semi-structured questionnaire (S1 File). The focal persons were individuals responsible for managing or coordinating cervical cancer services. However, the number of cervical cancer service providers was different from a health center to a health center.

### Data collection procedure

The following two components were part of this study's data collection process:

1. Face-to-face interviews: The data were collected from the 51 health centers through face-to-face interviews with cervical cancer focal persons by five trained data collectors (public health experts) using an Open Data Kit (ODK) tool installed with an English language semi-structured questionnaire (S1 File). The semi-structured questionnaire was read out to cervical cancer focal persons and explained (translated) in Amharic (local language) to avoid any misunderstanding. The survey was focused on health system "inputs" (providers, equipment, tests and vaccines, financing, and health information systems), and "outputs" (target achievement of services, availability of screening options, timeliness of services, patient-centeredness, and equity) of cervical cancer services.

2. Secondary data review: This involved a review of existing data related to cervical cancer. A checklist (for percentage achievement of public awareness, HPV vaccination, screening, and treatment) was used for the data extraction by examining records from the cervical cancer register, the extended program on immunization (EPI) register, and administrative reports. The cervical cancer register provided information on the number of women who were screened for cervical cancer and who received treatment. The EPI registration was checked to ascertain the number of girls who received the HPV vaccine. Administrative reports, on the other hand, provided information on public awareness messages related to cervical cancer.

## Study variables

To assess the health system inputs and outputs influencing access to cervical cancer prevention, screening, and treatment services, the study took into account a number of different variables. These indicators assisted in determining whether cervical cancer services were comprehensive, timely, patient-centered, and equitable to the population.

1. Health system inputs: The health system's "inputs" were determined by taking into account 1) Human resources: The availability of staff for services related to cervical cancer screening and treatment (nurses, midwives, and/or health officers); 2) Physical/Infrastructure: Availability of screening devices, diagnostic tests (HPV DNA tests and Pap smear tests), treatment equipment, and HPV vaccine; 3) Information: Health information systems that facilitate the collection, management, and analysis of data related to cervical cancer screening and treatment (cervical cancer register, EPI register, and administrative reports); and 4) Financial: Funding (infrastructure, supplies, and operational costs), user fees, and transportation costs.

2. Health system outputs: The measurement of health system "output" indicators to cervical cancer screening and treatment included 1) Target Achievement: Measuring the percentage of HPV vaccination, screening, and treatment achievement (performance per annual plan) towards meeting the WHO 90-70-90 cervical cancer targets [1]; 2) Availability: Assessing the availability of screening and testing (visual inspection with acetic acid, HPV DNA tests, and Pap smear tests); 3) Timeliness: Assessing the time required for screening and treatment services; 4) Patient-centeredness: Assessing whether patient preferences and needs were considered by providing various testing options; and 5) Equity: Evaluating the provision of cervical cancer prevention, screening, and treatment services without variation based on socioeconomic status, ability to pay, or geographical location.

## Data quality control

In this study, the quality of the data was controlled utilizing a variety of techniques. Five public health professionals were selected for data collection based on their prior experience with

quantitative data collection using the Open Data Kit (ODK) platform. Then, using the data collection guidelines, a half-day of training was given to the data collectors. The semi-structured questionnaire that was pre-installed on the ODK tool was the main topic of discussion. The data collectors offered comments on how to make the tool more effective at answering the study questions. The semi-structured questionnaire's validity was further evaluated based on how well it addressed the study questions through field testing. For this, the tool underwent pretesting at the Felege Meles Health Center in Addis Ababa before the actual data collecting began. The tool was modified in response to field feedback. Real-time data synchronization and collection were made accessible through the ODK central server. The questions pre-programmed on the ODK allowed for the validation of the tools' consistency, integrity, validity, and comprehensiveness. The data manager also supervised the daily ODK server data collection and consistency. Data auditing was done before data collection was complete to maintain the required sample size. Any discrepancies were fixed after determining the sample size before data analysis. At the same time, to guarantee the reliability of the study's findings, thorough data, and ongoing data comparisons were used.

## Data processing and analysis

To make sure that the data was correct, trustworthy, and prepared for descriptive analysis, the following approach was employed. Data auditing was done before data collection was finished to make sure the required sample size was kept. This process included checking the collected data for accuracy, consistency, and completeness. Any discrepancies or missing data points were fixed. The data was exported to Excel CSV format after the data auditing procedure was finished. To further clean and analyze the data, SPSS Version 26 was employed. At this step, extra verifications and checks were done on the data to make sure it was accurate and consistent. This made it possible to find mistakes and fix them, deal with missing values, and recode variable replies so that they are in the correct format. The dataset was cleaned and then made ready for additional analysis. In this step, certain variables of interest were chosen and the data was formatted in a way that is suitable for analysis. Descriptive statistics (frequency, proportion or percentage, mean, and standard deviation) were used to summarize the data. Graphs and tables were used to recap the percentage and attributes of the cervical cancer focal persons' responses, as well as the secondary data extracted from health centers' cervical cancer register, EPI register, and administrative reports.

## Ethical consideration

Ethical approvals of the study were obtained from 1) Strathmore University Institutional Scientific and Ethical Review Committee (SU-IERC1373/22), and 2) Addis Ababa City Administration Health Bureau Ethical Clearance Committee (A/A/0024/227). Support letters were also provided by 11 sub-cities in Addis Ababa and produced to the respective health centers for permission to collect data. The cervical cancer focal persons were briefed on the purpose and objective of the research. The written informed consent was obtained from respondents before the interview. The collected data was kept confidential, and no names were displayed in the study.

## Results

### Health system inputs/functions

The survey showed that 94.1% of the health centers had adequate staff for cervical cancer services. There was at least one nurse, one midwife, and one health officer trained in cervical

cancer management in 20, 17, and 13 health centers, respectively. Among the 51 health centers surveyed, the providers who received cervical cancer training in the 24 months preceding the study were 40 (78.4%) nurses, 38 (74.5%) midwives, 18 (35.3%) health officers, and 25 (49%) health extension workers (HEWs). All the fifty-one health centers were providing community services through HEWs. Public education was the main role for HEWs in 48 (94%) facilities, followed by referral linkage of women (92%). The HEWs public engagement areas were mostly in community outreach (100%) and at schools (71%). Table 1 shows speculums were available in 51 (100%), a cryotherapy machine with $CO_2$ gas supply in 50 (98%), acetic acid in 50 (98%), and the HPV vaccine in 28 (54.9%) facilities. The cervical cancer prevention and control guidelines were available throughout the fifty-one facilities, and the majority of them had adequate Information, Education, and Communication (IEC)/ Social and Behavior Change Communication (SBCC) materials including posters, flyers, and brochures. Audio and video messages were unavailable in 45 (88.2%) and 46 (90.2%) facilities, respectively. Paper-based referral forms were available throughout the health centers though only 26 (51%) of them received no feedback.

**Table 1. Availability of equipment, vaccines, and supplies at health centers in Addis Ababa (n = 51).**

| Variables | Count | Percent |
|---|---|---|
| Speculum | | |
| Available | 51 | 100.0 |
| Cryotherapy machine with $CO_2$ gas supply | | |
| Available | 50 | 98.0 |
| Not available | 1 | 2.0 |
| Acetic acid | | |
| Available | 50 | 98.0 |
| Not available | 1 | 2.0 |
| HPV vaccine | | |
| Available | 28 | 54.9 |
| Not available | 23 | 45.1 |
| IEC/SBCC material (poster) | | |
| Available | 47 | 92.2 |
| Not available | 4 | 7.8 |
| IEC/SBCC material (audio) | | |
| Available | 6 | 11.8 |
| Not available | 45 | 88.2 |
| IEC/SBCC material (video) | | |
| Available | 5 | 9.8 |
| Not available | 46 | 90.2 |
| IEC/SBCC material (flyer) | | |
| Available | 39 | 76.5 |
| Not available | 12 | 23.5 |
| IEC/SBCC material (brochure) | | |
| Available | 44 | 86.3 |
| Not available | 7 | 13.7 |
| Cervical cancer prevention and control guideline | | |
| Available | 51 | 100.0 |
| Referral forms | | |
| Available | 51 | 100.0 |

The cervical cancer screening and treatment services at public health centers were provided for free. It was covered by the government, development partners, and community-based health insurance funds. Only 33% of the fifty-one facilities reported a shortage of funds. There were no transportation incentives for the linkage of the most at-risk women to facilities. Thirty-three (64.7%) of the health centers indicated that there was inadequate transportation for cases that were referred to hospitals. Nevertheless, 39 (76.5%) of the facilities' distances to specialized care were determined to be less than 10 km. Patient follow-up telephone calls were practiced by 39 (76.5%) health centers, especially for re-screening testing. Cervical cancer screening and treatment, and EPI services registration books were available in all health centers.

## Health system outputs/performance

**Target achievement of services.** Messages on cervical cancer awareness were delivered to 79.1% of women, 61.2% of girls aged 15 were fully vaccinated against cervical cancer, 71.1% of women screened for cervical cancer, and 79.8% of women treated for precancer lesions (Table 2).

**Equity, availability, patient-centeredness, and timeliness of services.** Efforts have been made by the government to attain equitable free cervical cancer services through at least 90 active public health centers spread out in 11 sub-cities in Addis Ababa with no discrimination due to income, gender, place of residence, and ethnicity of users. This was demonstrated by the cervical screening, preventive, and treatment services offered in the 51 health centers examined. Since Pap smear tests were not available in health centers, women were referred for testing in private diagnostics or the Family Guidance Association of Ethiopia costing USD10 per test. The turnaround time (TAT) was ranging from 15 to 30 days in 49 (96.1%) of the health centers. Whereas the TAT for the visual inspection with the acetic acid test was 1 minute in 48 (94.1%) of the health centers. HPV DNA test was unavailable throughout the 51 health centers. During the January 2022 campaign, 35,099 vials of HPV vaccine were wasted due to contamination and breakage, and 4,240 vials of vaccine as a result of the Vaccine Vial Monitor (VVM) change. In 41 (80.4%) of the health centers, women had ongoing relationships with the healthcare providers and were given a chance to choose the providers in 33 (64.7%) of the facilities. The average waiting time for cervical cancer screening and treatment was 30 minutes or longer in 41% of facilities. E-registers were not available in the majority of the facilities (86.3%).

## Major challenges and improvement areas for cervical cancer services

The survey showed that 96.1% of health centers admitted to facing challenges in delivering public education with 58.8% mentioning inadequate media coverage as the major underlying factor. The women's transportation cost (35.3%) and the facility's distance (29.4%) were

**Table 2. Achievement (%) of cervical cancer services at health centers in Addis Ababa (n = 51).**

| Achievement* | Minimum | Maximum | Mean | Std. Deviation |
|---|---|---|---|---|
| % of women reached with cervical cancer awareness messages. | 20 | 99 | 79.1 | 22.4 |
| % of girls fully vaccinated against cervical cancer at the age of 15. | 0 | 99 | 61.2 | 25.6 |
| % of women screened for cervical cancer. | 13 | 99 | 71.1 | 24.3 |
| % of women with positive cervical screening treated. | 1 | 99 | 79.8 | 33.3 |

*Achievement: Performance per plan for July 1, 2021 to June 30, 2022.

**Table 3. Prevention, screening, and community linkage challenges, and improvement areas at health centers (n = 51).**

| Variables | Count | Percent |
|---|---|---|
| Prevention and screening service challenge* | | |
| Education | 49 | 96.1 |
| Religious factors | 7 | 13.7 |
| Stigma | 9 | 17.6 |
| Language barrier | 5 | 9.8 |
| Lack of space for screening | 10 | 19.6 |
| Unavailability of SBCC/IEC materials | 6 | 11.8 |
| Inadequate media coverage (TV, radio) | 30 | 58.8 |
| Major challenges on community linkage* | | |
| Transportation cost | 18 | 35.3 |
| Distance to the facility | 15 | 29.4 |
| Capacity of HEWs | 3 | 5.9 |
| Fear of procedure | 8 | 15.7 |
| Lack of awareness | 12 | 23.5 |
| Areas of improvement in cervical cancer services* | | |
| Provide transportation allowance | 11 | 21.6 |
| Shorten waiting time | 8 | 15.7 |
| Provide staff training | 41 | 80.4 |
| Improve diagnostic capacity | 31 | 60.8 |
| Ensure preference for a provider | 13 | 25.5 |
| Spouse or partner support | 21 | 41.2 |
| Promote media coverage | 39 | 76.5 |
| Facilitate suitable room | 7 | 13.7 |
| Include routine HPV vaccine service | 4 | 7.8 |

*Participants provided more than one response.

additional concerns. Training to providers was recommended by 80.4% of the cervical cancer focal persons with 76.5% suggesting improved media coverage (Table 3).

## Discussion

This study looked at the effect of the health system on access to cervical cancer services from the primary healthcare perspective. It is the first survey to examine the "inputs" and "outputs" of the health system regarding public health centers' response and dedication to the fight against cervical cancer. The study revealed the prevailing health system challenges deterring the provision of the highest possible coordinated and integrated cervical cancer services at the facilities.

### Health system inputs

In-service training had been given to 35% of health officers and 49% of health extension workers (HEWs) in the 24 months before the study. This supports previous studies conducted in Addis Ababa, Ethiopia, which found that more than half (52%) of HEWs lacked an appropriate understanding of cervical cancer and its screening [23, 24]. In addition, only 16% of health centers and hospitals' staff who provide services for cervical cancer in East Gojjam Zone,

northwest Ethiopia, had undergone in-service training [25]. However, our study revealed a better development than a previous study conducted in Ethiopia, which discovered that only 4% of the employees in the health facilities had undergone in-service training on cancer in 2014 [14]. All health centers possessed guidelines for preventing and managing cervical cancer. Again this showed significant improvement compared to a prior study in Ethiopia, which discovered that only 8% of healthcare facilities held cancer guidelines in 2014 [14]. The lack of audio materials in 88% and video recordings in 90% of facilities, may have had a detrimental impact on the public awareness achievement of 79% shown in this study. This confirms the limited community awareness identified by previous studies conducted in Ethiopia [12, 26], and other countries [27–29]. The decision-makers may take into account online learning modules to satisfy the current demands for in-service training. For a routine immunization campaign aimed at 14-year-old girls, the quadrivalent HPV vaccine was introduced in Ethiopia in December 2018 [30]. However, this was given in campaigns and the HPV vaccine was available in 54.9% health centers only. In 35% of health centers, the cost of transportation was a significant barrier to connecting women in the community with services for cervical cancer. This probably contributed to Ethiopia's previously reported low cervical cancer screening service utilization of 5.47% [15]. Lessons from the HIV programs may also be applied to the design and implementation of transportation incentives, particularly for connecting the most vulnerable women to health centers.

## Health system outputs

In comparison to the annual plan for 2022, 61% of girls received the full HPV vaccination by the age of 15 to fulfill the WHO target of 90% by 2030 [1]. Only 29.4% of health centers' HPV immunization programs involved HEWs. Additionally, the survey found high HPV vaccine loss (39, 339 vials). To advance towards reaching the WHO target of 90% by 2030, further efforts were needed through routine immunization programs, effective supply chain management, and improved HEWs participation. In comparison to the WHO goal of 70% by 2030, a higher screening success rate (71%) has been attained. Per the cervical cancer guidelines algorithm, the turnaround time (TAT) for counseling, screening, and treating women was less than 15 minutes [30]. However, the health centers had no technical capacity and infrastructure, and poor access to Pap smear testing. It was performed outside the facilities with a TAT of up to a month and out-of-pocket charges. These findings uphold the previous Ethiopian studies that reported inadequate diagnostic and treatment centers for cancer [12, 13]. Furthermore, three-fourths of the patients had delayed diagnostic confirmation (more than 30 days) in a different study conducted in Addis Ababa, Ethiopia [31]. Previous research suggested combining a Pap smear test with visual inspection using acetic acid (VIA) to boost the success rate of cervical cancer screening [32, 33]. The challenges encountered with Pap smear testing could be lessened with proper lab networking, test cost assistance, and a shorter TAT. In addition, implementing Pap smear testing in facilities could enhance screening and lead to better achievement for treating precancer lesions than what this study found (80%).

To reduce long wait times, health facility managers could have set timelines and monitored the services from triage to screening clinics with improved provider preference assurances. Offering accurate, thorough, and standardized patient condition information was difficult because most health centers (86%) still needed electronic health records (EHRs). Moreover, clinical records of patients were not electronically referred from health centers to hospitals and vice versa. Though all health centers managed paper-based referrals no feedback was received in 51% of the health centers which demanded a health system practice redesign to access to the patient's clinical records. According to a Tanzanian study, similar issues were encountered in

this regard [28]. A robust information system is essential for enabling the integration and coordination of patient-centered care as well as round-the-clock provider access [26, 34]. Furthermore, this may facilitate regular evaluation of patient-reported indicators to determine whether patient-centered care is being provided and to fix any gaps in the health system [35].

## Equity of services

Equity of services has been given top priority in the Ethiopian Health Sector Transformation Plan II [19]. It places a strong emphasis on making sure that nobody is prevented from accessing essential healthcare because of their location or other traits, such as being a woman or a member of a disadvantaged group. The expansion of equitable health services at health centers was one of the main strategies of the health transformation plan. As a result, throughout eleven sub-cities of Addis Ababa, the Addis Ababa City Administration Health Bureau has made notable strides in guaranteeing the availability and accessibility of cervical cancer screening and treatment services in at least ninety health centers. Our study showed that services for cervical cancer prevention and control were offered throughout the 51 health centers. By achieving equity, Addis Ababa may be able to ensure that everyone has an equal opportunity to gain from cervical cancer prevention, early detection, and treatment. On the other hand, the replies from 76.5% of the health centers indicated that women had to travel up to 10 kilometers for specialized care, which is only provided in secondary and tertiary hospitals. According to a study from Zimbabwe, women should be provided with transportation to health facilities so they can undergo cervical cancer treatment [27].

## Areas of improvement

Training for healthcare providers, spreading messages of public awareness, and improving diagnostic capabilities at health centers were all highlighted as key health system deficiencies that need to be addressed. These priorities were also identified by several earlier research [12, 14, 25–27, 29, 36]. Prior research suggested coordinated efforts to lower barriers to screening services, primarily by utilizing the media to create demand and offering training to healthcare providers [26, 37, 38]. Other studies conducted in Ethiopia and Central America proposed improving HPV DNA testing through the self-collection of samples to enhance screening in high-burden and resource-limited settings [39, 40]. Furthermore, a study carried out in Jimma, southwest Ethiopia, suggested that initiatives be taken to enhance women's views, comprehension, and satisfaction regarding cervical cancer screening services [41].

## Strengths and limitations of the study

The study has a variety of benefits regarding the health system's influence on the provision of services for cervical cancer prevention and control in settings with limited resources. It offers insightful information about the "inputs" and "outputs" of the healthcare system about cervical cancer services. However, the study also revealed several restrictions that need to be taken into account. The inability to evaluate the aspects of the health system that have an impact on the provision of cervical cancer services at private health institutions is the first restriction. Since the study concentrated on situations with few resources, private health centers may have a big impact on how cervical cancer services are provided. The absence of these facilities could preclude the study from giving a complete picture of the health system and its impact on cervical cancer services. Another restriction is the generalizability of the study's findings. Since the study was conducted in Addis Ababa, its findings might apply to major urban settings, but they might not fairly represent the challenges that other regions and rural parts of the country face. When extrapolating the results beyond the study's confines, care should be used because

different places may have different infrastructures and access to services. Last but not least, the study did not completely address the issues with Pap smear testing's technical capabilities and infrastructure, nor did not look into any missed chances with the HPV vaccination program. Comprehending the complete range of cervical cancer preventive and control services requires an understanding of these factors. However, due to budget limitations, the study was unable to investigate these areas, which restricts the findings' comprehensiveness. It is crucial to be aware of these limitations when evaluating the study's results and choosing how to move forward with cervical cancer prevention and control activities in settings with limited resources. A more nuanced understanding of the aspects of the health system that have an impact on the delivery of cervical cancer services could be provided by additional research that takes these constraints into account and contributes to the development of useful reform measures.

## Implications for policy and practice

The primary aim of the National Cancer Control Plan (NCCP) is to engage the entire health system, including primary healthcare facilities such as health centers. It is the responsibility of the decision-makers at the national and regional levels to monitor the entire system, pinpoint its flaws, and create plans to enhance health performance outcomes. While decision-makers concentrate on the NCCP's overarching objectives, healthcare professionals working in health centers are dedicated to providing patient-centered integrated and coordinated standard cervical cancer care that caters to the requirements of both individual service users and communities. This study looked at how health centers responded to and were committed to the NCCP's overarching goals by evaluating the current service provision, organizational capacity, information, and patient participation. It highlighted weaknesses in the healthcare system, including low rates of HPV vaccination, a lack of screening choices caused by the sporadic availability of HPV DNA tests, a lack of access to Pap smear tests, insufficient staff capacity building, and lengthy wait times. The results of the study could potentially provide a foundation for tackling the challenges that are encountered in other forms of cancer. This may spur additional studies to look into and fill in these gaps so that everyone afflicted by cancer can receive thorough and patient-centered care. Researchers can contribute to a deeper understanding of the health system's strengths and shortcomings in providing effective cancer prevention, treatment, and control measures by extending the scope of their research beyond cervical cancer and investigating the execution of NCCP goals for other cancers. Lastly, this research can support policy initiatives by directing focus, establishing priorities, and revamping the nation's cancer care delivery system.

## Conclusions

To deal with cancer, the Ethiopian government has put the National Cancer Control Plan into effect. Free and equitable cervical cancer screening and treatment services were provided in over 90 health centers located throughout eleven Addis Ababa sub-cities. The programs, which were primarily aimed at women in the 30 to 49 years old age range, were bolstered by public awareness messages. Additionally, campaigns were used to immunize females against HPV. However, there were still challenges in the way of achieving higher HPV vaccination rates, which call for consistent access to vaccines with minimal wastes. Moreover, there were gaps in training for healthcare professionals, offering testing options and provider preferences, reducing wait times, and implementing electronic health records, and raising awareness campaigns primarily via audio and video messages, and television media. Addressing the inconsistent accessibility of HPV DNA testing could enhance screening services [39, 40]. The adoption of electronic health records is the primary means of enhancing the referral pathways between

health centers and hospitals to better coordinate and integrate services. Reducing wait times, enhancing public awareness, and fulfilling patient needs and preferences may be achieved by increasing providers' capacity and improving service coordination at facilities. The WHO 90-70-90 targets for cervical cancer can be met by 2030 with the provision of high-quality services at healthcare institutions and the combined efforts of communities, service users, and decision-makers.

## Supporting information

**S1 File. Semi-structured questionnaire.**
(DOCX)

**S2 File. Health centers survey tables and figures.**
(DOCX)

**S3 File. Health centers survey data_PLoS.**
(SAV)

## Acknowledgments

We are grateful to the Addis Ababa City Administration Health Bureau for approving the study, as well as the 11 Addis Ababa sub-cities for facilitating access, providing letters of support to the health centers, and offering ongoing assistance during the study. Additionally, we would like to thank all of the study participants, particularly the cervical cancer focal persons who supplied pertinent data. Furthermore, we would like to express our sincere gratitude to all data collectors who helped with the validation of the study's questionnaire and data collection.

## Author Contributions

**Conceptualization:** Kemal Hussein, Gilbert Kokwaro, Francis Wafula, Getnet Mitike Kassie.

**Data curation:** Kemal Hussein.

**Formal analysis:** Kemal Hussein.

**Investigation:** Kemal Hussein.

**Methodology:** Kemal Hussein, Gilbert Kokwaro, Francis Wafula, Getnet Mitike Kassie.

**Supervision:** Gilbert Kokwaro, Francis Wafula, Getnet Mitike Kassie.

**Validation:** Kemal Hussein, Gilbert Kokwaro, Francis Wafula, Getnet Mitike Kassie.

**Writing – original draft:** Kemal Hussein.

**Writing – review & editing:** Gilbert Kokwaro, Francis Wafula, Getnet Mitike Kassie.

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
