## [Decision Letter · Decision Letter 0]

3 Aug 2023

PONE-D-23-08979Health system factors affecting equitable access to quality cervical cancer services at public health centers in Addis Ababa, EthiopiaPLOS ONE

Dear Dr. Hussein,

Thank you for submitting your manuscript to PLOS ONE. After careful consideration, we feel that it has merit but does not fully meet PLOS ONE’s publication criteria as it currently stands. Up on my own review and the reviewers comments I recommend Major Revision to this manuscript. Therefore, we invite you to submit a revised version of the manuscript that addresses the points raised during the review process. Please pay attention to editor and reviewers comments below.

ACADEMIC EDITOR: Please pay attention to the following points in addition to feedback given by reviewers:There are major flaws in grammar and language throughout this paper. In the revised version, try to address this. Authors are highly recommended to get editions services of this manuscript with experts, to have a qualified scientific paper. The abstract section of the paper  should clearly show the rationale for this paper, and how the proposed aim was answered. But, this issues are shallow and lacks clarity. The method section in the main body is not also clearer and doesn't help for replication. Therefore,  authors are suggested to revise this section and better if the methods is re-organized in the follows sections: study settings and period, Study design and participants, data collection,  study variables,  data quality control, data processing and analysis,  ethical consideration In the topics and aim of the paper, key terms like equity, quality of cervical cancer services and the like are described.  However, authors didn't adequately define this terms in any section of the paper,  not described the measurement methods employed, and displayed the measured results.  Though two frameworks are employed in the current paper, how this framework employed to guide the measurements lacks clarity. Please provide explanations to this, and as well address this issues within the manuscript. Make sure that the manuscript has addressed the journals requirements in terms of contents, particularly after the conclusions section of the paper. Refer the journal guideline and correct this part. Change the citation formats to square brackets (e.g., [1]).Adequately discuss the present study findings with that of literature in other settings(out of Ethiopia). Discuss the strength and limitations of the study, focusing on the methods used, and have a separate section to discuss the policy and practice implications of the study. ==============================

We look forward to receiving your revised manuscript.

Kind regards,

Dawit Wolde Daka

Academic Editor

PLOS ONE

Journal Requirements:

Reviewers' comments:

Reviewer's Responses to Questions

**Comments to the Author**

1. Is the manuscript technically sound, and do the data support the conclusions?

Reviewer #1: Partly

Reviewer #2: Yes

2. Has the statistical analysis been performed appropriately and rigorously? 

Reviewer #1: No

Reviewer #2: Yes

3. Have the authors made all data underlying the findings in their manuscript fully available?

Reviewer #1: Yes

Reviewer #2: Yes

4. Is the manuscript presented in an intelligible fashion and written in standard English?

Reviewer #1: Yes

Reviewer #2: Yes

5. Review Comments to the Author

Reviewer #1: Manuscript Number: PONE-D-23-08979

Manuscript Title: Health system factors affecting equitable access to quality cervical cancer services at public health centers in Addis Ababa, Ethiopia:

Dear Editor,

Thank you for inviting me to review the scientific quality of this article.

Dear Authors,

It is very interesting to get research done on this very interesting area of women's health.

However, I recommend you the following points be addressed before your article gets published

1. Either modify the title or re-do the analysis because there is no result to talk about the quality of service. Also, there is no strongly concluded point about equity to the service user.

2. The 150,000 new cases that Ethiopia reports is not clear. Line 46

3. Font Color of lines 56-59 is different: a light blue. Better to make it similar throughout the document. The sentence is direct copy and paste. Rephrase

4. I was expecting to read what “quality cervical cancer services” are mean in the introduction, particularly in Ethiopia. How did it become your concern? Any other previous evidence on the case and what is the gap?

Material and Method

1. The method needs rearrangement. For example, why data collection method under the session “Study design and period”?

2. Organization of Ethiopian health service delivery system line 93-95 needs reference.

3. It is described how the health facilities were sampled and selected. How about the service providers? Who were they? How were they selected? It should not be indicated under the session “Data collection procedure and data analysis”

4. The data collection procedure is not indicated well. Face-to-face interview? FGD? Checklist data extraction???

5. Avoid unnecessary repetitions. For example, the study period on lines 91 and 118.

6. The data analysis procedure lacks a detailed explanation. Is it not important to explain the used analysis method, and cutoff points,…the analysis is rough and do not fully support the study aims.

7. It is important to describe measurements in this study. For example,

o How the fulfillment of the input was measured? Was it the summation of all the available inputs or any of them? Was 100% input expected?

o What indicators were used to measure (effectiveness, availability, timeliness, and patient-centeredness): the Outputs?

o How was equity of cervical cancer services measured? When did you say there is equity in the services?

o What is quality service as per your study?

Result

1. What is the importance of writing in different colors again? Line 134-139

2. You have no data extraction method. How was the data for the information provided in Table 3 were obtained?

Discussion

1. Lines 192-194: “The study revealed that the prevailing health system challenges at the primary healthcare level had negatively impacted the quality of cervical cancer 194 services.” How can you say it affects the quality without showing the status of the quality of the service?

2. Ideas in lines 197-216 seem taken from literature; how does the idea on line 217 connect with it? Which findings are being compared?

3. Line 233-234, client report. Where did you get it? You have no client involvement or data extraction in the data collection method.

4. In general, I recommend doing the discussion ideally under sub-headings of input, output, and equity in access to cervical cancer screening. It is a copy of the result. Make it short and discuss using guidelines on recommended minimum importance.

Conclusions

Lines 312-313: do you have data on the poor engagement of school and drop-out girls?

Similarly, in lines 316-317; do you have information on “HPV DNA and Pap smear testing attract more women for screening”?

the conclusion should be done depending only on the data that can be accessed in this manuscript. It seems discussion.

Reviewer #2: General Comments: Thank you for the updated manuscript. Any study on equitable access to quality cervical cancer services in LMIC countries is important as efforts are made to control/eliminate CC globally from the perspective of health system factors. The manuscript has paramount but needs modification to make it publishable.

I have pointed some out below but get this fixed throughout the manuscript.

Specific comments

Abstract

Lines 19-20: be consistent in the use of terms (women vs girls) use either of the two throughout your documents, again in the use of health center and health facilities

Introduction

1. Line 39: add references at the end of the paragraph.

2. Line 56-59: be alert while taking documents from other sources and I will recommend you to re-write again.

Methods

3. Lines 92 – 101: what is your study population: healthcare providers or health centers??

4. A proportionate number of 51 facilities were randomly selected using Microsoft Excel out of a total of 90 public health centers actively providing cervical cancer services. How do you select healthcare providers from each health center? Do you think equal numbers of healthcare providers were in each HC?

5. Line 113-114: you said cervical cancer service providers in facilities (11 nurses, 10 midwives, 4 health officers, and 26 cervical cancer focal persons) were approached

What is your justification to select nurses, midwifery, and health officers? Why not used cervical cancer focal persons throughout all health facilities

6. Line 114: add health be for the word facilities, again be consistent in the use of health centers and facilities( I recommend using health facilities throughout your documents)

7. You need to provide more details on the study recruitment process and data collection.

o How exactly were the healthcare providers recruited from 51 health centers?

o How do you select 51 health centers from 11 sub-cities? Clearly shown in sampling procedures

o Where was the data collected, and for how long? Etc. etc.

o Since these healthcare providers were not interviewed in the local language even if they were understanding English? What measures were in place to ensure that the translation process, did not affect the content of what was said?

o Lines 120: change ethical consideration to Ethics approval and consent to participate. I recommend you strictly follow journal guidelines

o Lines 125-126: All methods were performed in accordance with the relevant guidelines and regulations (Declaration of Helsinki). Do the PLoS One guidelines follow such terms?

o On line 174: table 3: I think you have used document review for data retrieval, my concern is what are your data collection methods( state clearly)

Results

General comments

If the socio-demographic variables were in your tools, better to add the description of socio-demographic characteristics of health care providers ( age, sex, profession, etc)

8. Lines 130-131: your description were 20, 17, and 13 health facilities. The total was 50 versus 51 health facilities.

9. Lines 131-132: Rephrase the sentence at lines 131 and 132 to make correct meaning

The recommendation is to minimize use respectively when your list of activities were more than three. In 40 (78.4%), 38 (74.5%), 18 132 (35.3%), and 25 (49%) facilities had nurses, midwives, health officers, and health extension 133 workers (HEWs) who received cervical cancer training, respectively

You can write as 40 (78.4%) nurses, 38 (74.5%) midwives, 18 (35.3%) health officers, and 25 (49%) health extension workers (HEWs) received cervical cancer training

10. Line: 142: table 1: revised all tables based on journal guidelines formats

11. Lines 161-165: this whole text descriptions were not in line with the tables and need to be corrected

12. Lines 170: How do you measure the effectiveness of services? needs operational definition in the methods part

Discussion and conclusions

Detail and clear description of each section

6. PLOS authors have the option to publish the peer review history of their article (what does this mean?). If published, this will include your full peer review and any attached files.

Reviewer #1: No

Reviewer #2: No

---

## [Author Response · Author response to Decision Letter 0]

16 Sep 2023

A rebuttal letter that responds to each point raised by the academic editor and reviewer(s) has been uploaded as a separate file labeled 'Response to Reviewers'.

---

## [Decision Letter · Decision Letter 1]

15 Nov 2023

PONE-D-23-08979R1Health system factors affecting access to cervical cancer prevention, screening, and treatment services at public health centers in Addis Ababa, Ethiopia.PLOS ONE

Dear Dr. Hussein,

Thank you for submitting your manuscript to PLOS ONE. After careful consideration, we feel that it has merit but does not fully meet PLOS ONE’s publication criteria as it currently stands. Therefore, we invite you to submit a revised version of the manuscript that addresses the points raised during the review process.

Kind regards,

Dawit Wolde Daka

Academic Editor

PLOS ONE

Journal Requirements:

Reviewers' comments:

Reviewer's Responses to Questions

**Comments to the Author**

1. If the authors have adequately addressed your comments raised in a previous round of review and you feel that this manuscript is now acceptable for publication, you may indicate that here to bypass the “Comments to the Author” section, enter your conflict of interest statement in the “Confidential to Editor” section, and submit your "Accept" recommendation.

Reviewer #2: All comments have been addressed

Reviewer #3: (No Response)

2. Is the manuscript technically sound, and do the data support the conclusions?

Reviewer #2: Yes

Reviewer #3: Yes

3. Has the statistical analysis been performed appropriately and rigorously? 

Reviewer #2: Yes

Reviewer #3: Yes

4. Have the authors made all data underlying the findings in their manuscript fully available?

Reviewer #2: Yes

Reviewer #3: Yes

5. Is the manuscript presented in an intelligible fashion and written in standard English?

Reviewer #2: Yes

Reviewer #3: No

6. Review Comments to the Author

Reviewer #2: All my concerns were clearly stated and responded. Still your conclusion was so long and seems result , try to revised it.

Reviewer #3: (No Response)

7. PLOS authors have the option to publish the peer review history of their article (what does this mean?). If published, this will include your full peer review and any attached files.

Reviewer #2: No

Reviewer #3: No

---

## [Author Response · Author response to Decision Letter 1]

13 Dec 2023

Response to reviewers: A rebuttal letter that responds to each point raised by the reviewers has been uploaded as a separate file labeled 'Response to Reviewers'.

---

## [Decision Letter · Decision Letter 2]

22 Feb 2024

Assessing the influence of the health system on access to cervical cancer prevention, screening, and treatment services at public health centers in Addis Ababa, Ethiopia.

PONE-D-23-08979R2

Dear Dr. Hussein,

We’re pleased to inform you that your manuscript has been judged scientifically suitable for publication and will be formally accepted for publication once it meets all outstanding technical requirements.

Kind regards,

Dawit Wolde Daka

Academic Editor

PLOS ONE

Additional Editor Comments (optional):

Reviewers' comments:

Reviewer's Responses to Questions

**Comments to the Author**

1. If the authors have adequately addressed your comments raised in a previous round of review and you feel that this manuscript is now acceptable for publication, you may indicate that here to bypass the “Comments to the Author” section, enter your conflict of interest statement in the “Confidential to Editor” section, and submit your "Accept" recommendation.

Reviewer #2: All comments have been addressed

2. Is the manuscript technically sound, and do the data support the conclusions?

Reviewer #2: Partly

3. Has the statistical analysis been performed appropriately and rigorously? 

Reviewer #2: Yes

4. Have the authors made all data underlying the findings in their manuscript fully available?

Reviewer #2: Yes

5. Is the manuscript presented in an intelligible fashion and written in standard English?

Reviewer #2: No

6. Review Comments to the Author

Reviewer #2: The author will be alert during revision. Some part of the manuscript were totally changed and finally come to unclear concepts.

7. PLOS authors have the option to publish the peer review history of their article (what does this mean?). If published, this will include your full peer review and any attached files.

Reviewer #2: No

---

## [Editor Report · Acceptance letter]

7 May 2024

PONE-D-23-08979R2 

PLOS ONE

Dear Dr. Hussein, 

I'm pleased to inform you that your manuscript has been deemed suitable for publication in PLOS ONE. Congratulations! Your manuscript is now being handed over to our production team.

Kind regards, 

on behalf of

Mr Dawit Wolde Daka 

Academic Editor

PLOS ONE